# Exploring the Spatial Variation Characteristics and Influencing Factors of PM$_{2.5}$ Pollution in China: Evidence from 289 Chinese Cities

**Shen Zhao [1,2] and Yong Xu [1,2,*]**

[1]   Institute of Geographic Sciences and Natural Resources Research, Chinese Academy of Sciences, Beijing 100101, China
[2]   Department of Geography, University of Chinese Academy of Sciences, Beijing 100049, China
*   Correspondence: xuy@igsnrr.ac.cn

**Abstract:** Haze pollution has become an urgent environmental problem due to its impact on the environment as well as human health. PM$_{2.5}$ is one of the core pollutants which cause haze pollution in China. Existing studies have rarely taken a comprehensive view of natural environmental conditions and socio-economic factors to figure out the cause and diffusion mechanism of PM$_{2.5}$ pollution. This paper selected both natural environmental conditions (precipitation (PRE), wind speed (WIN), and terrain relief (TR)) and socio-economic factors (human activity intensity of land surface (HAILS), the secondary industry's proportion (SEC), and the total particulate matter emissions of motor vehicles (VE)) to analyze the effects on the spatial variation of PM$_{2.5}$ concentrations. Based on the spatial panel data of 289 cities in China in 2015, we used spatial statistical methods to visually describe the spatial distribution characteristics of PM$_{2.5}$ pollution; secondly, the spatial agglomeration state of PM$_{2.5}$ pollution was characterized by Moran's $I$; finally, several regression models were used to quantitatively analyze the correlation between PM$_{2.5}$ pollution and the selected explanatory variables. Results from this paper confirm that in 2015, most cities in China suffered from severe PM$_{2.5}$ pollution, and only 17.6% of the sample cities were up to standard. The spatial agglomeration characteristics of PM$_{2.5}$ pollution in China were particularly significant in the Beijing–Tianjin–Hebei region. Results from the global regression models suggest that WIN exerts the most significant effects on decreasing PM$_{2.5}$ concentration ($p < 0.01$), while VE is the most critical driver of increasing PM$_{2.5}$ concentration ($p < 0.01$). Results from the local regression model show reliable evidence that the relation between PM$_{2.5}$ concentrations and the explanatory variables varied differently over space. VE is the most critical factor that influences PM$_{2.5}$ concentrations, which means controlling motor vehicle pollutant emissions is an effective measure to reduce PM$_{2.5}$ pollution in Chinese cities.

**Keywords:** PM$_{2.5}$ concentration; spatial variation; natural environmental conditions; socio-economic factors; China

## 1. Introduction

Numerous regions all over the world, especially developing countries such as China, are suffering from serious air pollution, which have attracted continuous concern in recent decades [1–4]. Since the reform and opening-up policy in 1978, China's total economic output has leaped to the second place globally; at the same time, China has been suffering from serious air pollution, of which haze pollution is of particular concern to the public [5,6]. Since 2013, the complex haze pollution, containing fine particulate matter (PM$_{2.5}$; $\leq 2.5$ μm in aerodynamic diameter) and coarse particulate matter (PM$_{2.5-10}$; 2.5 μm-10 μm in aerodynamic diameter) as core pollutants [7–9], has erupted in large-scale and long-term concentrations in many regions of China. It seriously affects the quality of China's economic

growth, public health, and government image. Compared with $PM_{2.5-10}$, $PM_{2.5}$ has stronger activity, a smaller particle size, and it is easier to carry these harmful substances deep into the respiratory tract, resulting in a significant increase in the risk of cardiac and pulmonary concerns and other related organs [8]. In addition, $PM_{2.5}$ has the ability to be transported over long distances and compared with other particles, remains airborne for a greater length of time [10,11]. Recent research has shown substantial evidence on the damage of $PM_{2.5}$ pollution to residents' health in many countries and regions [12–16].

Many researchers have been working to explore the causes and diffusion mechanism of $PM_{2.5}$ pollution [17–20]. Socio-economic factors (such as traditional biomass burning, motor vehicle exhaust emissions, construction dust, and so forth) have been proved to be closely related to the cause of $PM_{2.5}$ pollution [21–25].On the other hand, other studies have shown that natural environmental conditions (such as precipitation, wind speed, relative humidity, elevation, and so forth) are the critical factors affecting the diffusion of $PM_{2.5}$ pollution [26–30]. However, there are insufficient studies that explore the cause and diffusion mechanism of $PM_{2.5}$ pollution utilizing natural environmental conditions and socio-economic factors comprehensively. In addition, in previous studies, there were not enough sample cities to effectively depict the $PM_{2.5}$ pollution situation in China as a whole [31–33].

Both global regression models (including the ordinary least square (OLS) model, the spatial error model (SEM), and the spatial lag model (SLM)) and a local regression model (the geographic weighted regression (GWR) model) were utilized in this paper. The SEM model and SLM model were used to compensate for the defect that the OLS model fails to consider the spatial autocorrelation of dependent variables [34–36]. The GWR model was employed to simulate the spatial relationship between $PM_{2.5}$ concentration and its driving factors, which failed to be revealed in the three global regression models [37–39]. The GWR model captures spatial variations through allowing the parameters to change over space, for which reason it has been applied in many studies [40–42]. Specific explanations for the global and local regression models used in this paper are detailed in Section 2.

The main objectives of this paper are as follows: (1) to quantitatively describe the spatial variation of $PM_{2.5}$ concentrations and the six selected explanatory variables in all 289 sample cities; (2) to quantitatively identify the spatial variation relationship between $PM_{2.5}$ concentrations and the six selected explanatory variables; and (3) to quantitatively detect what is the most powerful driver of the spatial variation of $PM_{2.5}$ concentrations utilizing global and local regression models from both natural environmental conditions and socio-economic factors.

Our contributions lie in at least four aspects. First, the $PM_{2.5}$ spatial variation trend can be fully explained by taking into account the associated natural environmental conditions and socio-economic factors; second, the selected sample cities are sufficient and basically cover all typical regions of China; third, novel explanatory variables are selected, which provides a new perspective for studying the influencing factors related to $PM_{2.5}$ pollution; fourth, the $PM_{2.5}$ observation data and meteorological data used in this paper are sufficiently accurate (the original data is accurate to the hour level), which provides a good guarantee for the accuracy of the conclusions.

## 2. Data Sources and Methods

### 2.1. The Study Area

Figure 1 shows that the study area consists of 289 cities throughout China, which basically covers all the important cities in China. These cities were chosen on the condition of whether $PM_{2.5}$ monitoring points existed in the city and the availability of $PM_{2.5}$ data.

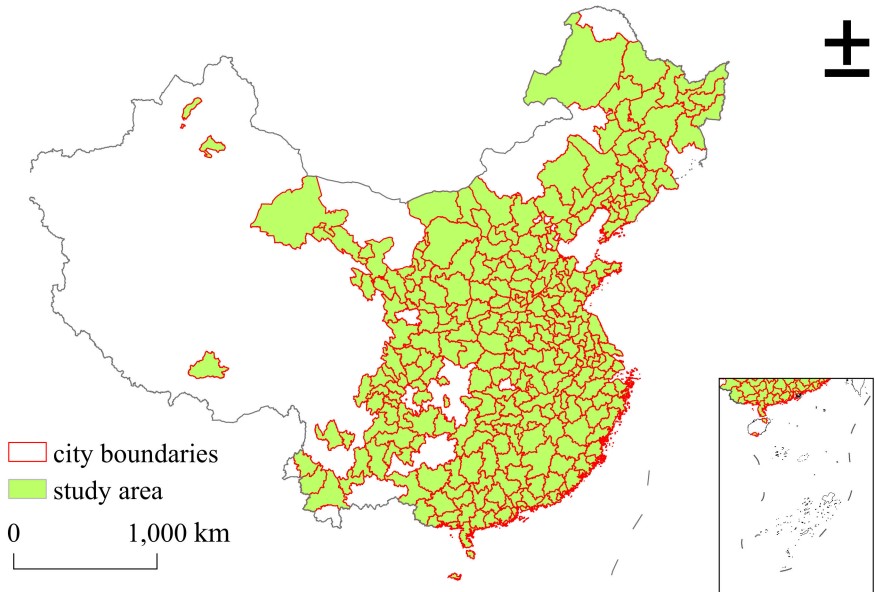

**Figure 1.** Distribution of the study area.

## 2.2. The Selected Explanatory Variables and Data Sources

This study explores the impacts of natural environmental conditions and socio-economic factors related to $PM_{2.5}$ pollution. In this section, the details of each explanatory variable are displayed (Table 1). Specifically, we selected precipitation (PRE, mm), terrain relief (TR) and wind speed (WIN, m/s) as the natural environmental conditions; while human activity intensity of land surface (HAILS, %), the secondary industry's proportion (SEC, %), and the total particulate matter emissions of motor vehicles (VE, $10^4$ tons) were chosen as the socio-economic factors.

**Table 1.** The selected natural environmental conditions and socio-economic factors related to particulate matter ($PM_{2.5}$) concentrations.

| Variables | Abbreviation | Meaning |
| --- | --- | --- |
| Terrain relief | TR | Relief degree of topography |
| Precipitation (mm) | PRE | Sum of daily precipitation in the year 2015 |
| Wind speed (m/s) | WIN | Mean of daily average temperature |
| Human activity intensity of land surface (%) | HAILS | The extent of the impact of human activities on land surface |
| The secondary industry's proportion (%) | SEC | The proportion of the secondary industry in the total social output value |
| The total particulate matter emissions of motor vehicles ($10^4$ tons) | VE | Total particulate matter emissions from motor vehicle exhaust |

The $PM_{2.5}$ pollution data in 2015 comes from the National Urban Air Quality Online Monitoring Platform [43], which contains data from 1497 air pollution observation stations (Figure 2a). The $PM_{2.5}$ data was carried out in accordance with the standards stipulated by the rules of Ministry of National Ecological Environment Protection of China strictly [44]. In order to be consistent with the standard terminology of GB3095-2012 [45], the daily average in this paper represented the arithmetic average of the average concentration in 24 hours of a natural day, the monthly average represented the arithmetic average of the average concentration in one calendar month, and the annual average represented the arithmetic average of the average concentration in one calendar year. The daily average data was calculated based on the hourly monitoring data of each station, and then based on the daily average data, the monthly data was measured. Finally, the annual average $PM_{2.5}$ concentrations of 1497 stations were calculated according to the monthly average, and then these values were interpolated in space to

obtain the annual average $PM_{2.5}$ concentrations of the selected cities. The meteorological data in 2015 came from the Resource and Environment Data Cloud Platform of the Institute of Geographic Sciences and Resources, Chinese Academy of Sciences (http://www.resdc.cn), which covered the data of all 2421 meteorological monitoring stations in China (Figure 2b). A spatial interpolation method (Inverse Distance Weighted (IDW)) was used to generate a national continuous surface for the meteorological data (including annual precipitation and annual average wind speed), and then the spatial statistical method was used to process the data to obtain the meteorological data of the 289 selected cities. The SEC data and the VE data in the 289 selected cities in 2015 were all from the statistical yearbook provided by the China Economic and Social Big Data Research Platform (http://data.cnki.net/). The HAILS data was calculated based on the algorithm proposed by Xu et al. [46], and the original data came from the China Statistical Yearbook for County Areas (2015) (http://data.cnki.net/). The TR data was calculated from the Digital Elevation Model (DEM), which came from the Cold and Arid Science Data Center (http://westdc.westgis.ac.cn).

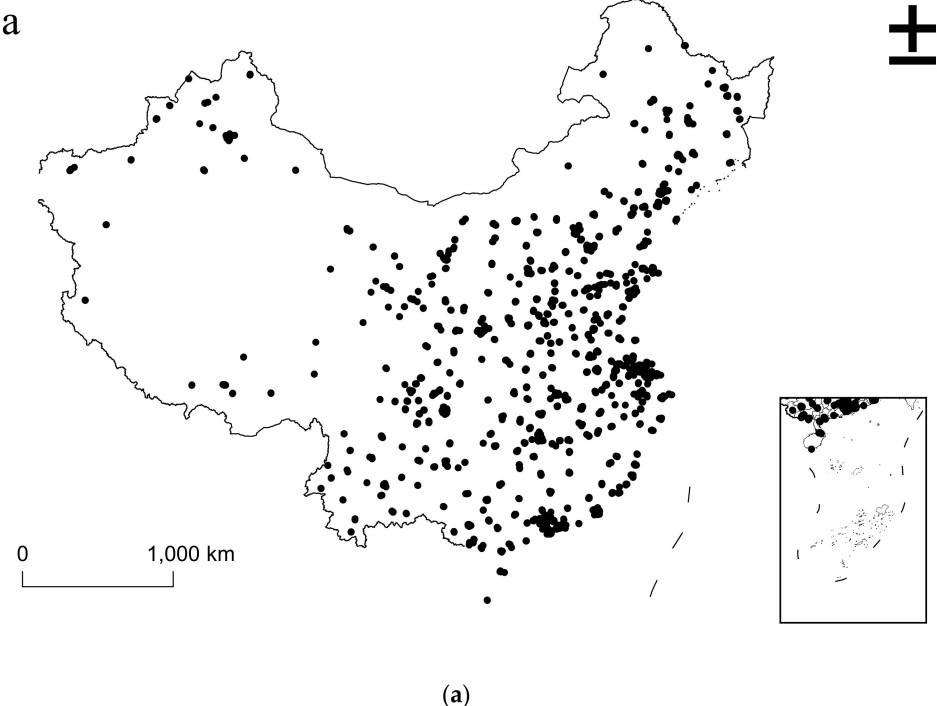

(**a**)

**Figure 2.** *Cont.*

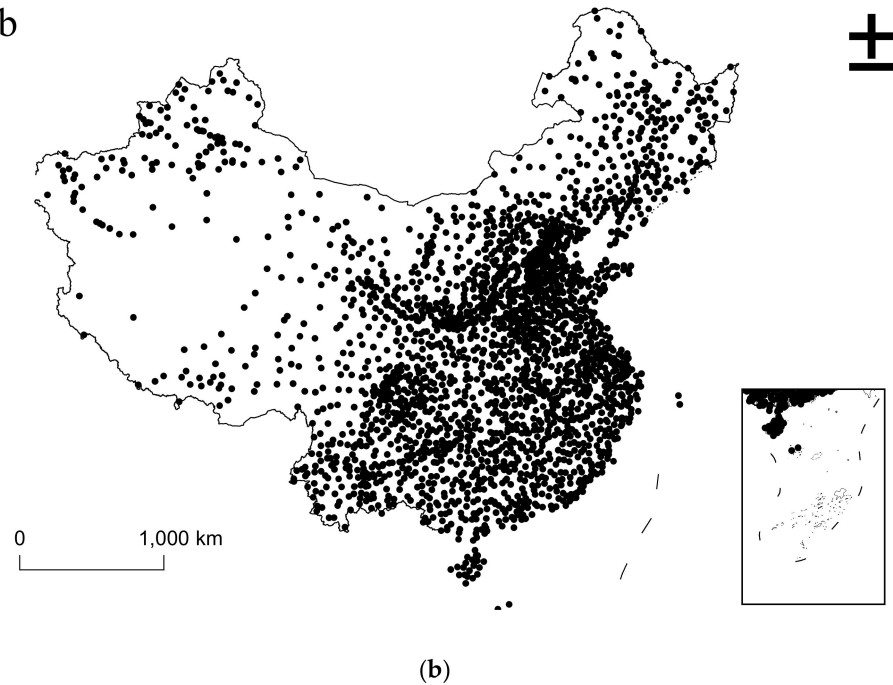

(**b**)

**Figure 2.** The locations of (**a**) 1497 air pollution observation stations and (**b**) 2421 meteorological monitoring stations.

## 2.3. Methods

This article first used Moran's *I* to detect the spatial autocorrelation of PM$_{2.5}$ pollution in the 289 selected cities [47–49], and subsequently used regression analysis methods to detect the relationship between PM$_{2.5}$ pollution and the selected explanatory variables.

### 2.3.1. Spatial Autocorrelation Analysis Based on Moran's I

In this paper, we used OpenGeoDa 1.12.1 software (Center for Spatial Data Science, University of Chicago, Chicago, Illinois https://www.uchicago.edu/research/center/center_for_spatial_data_science/) to achieve the spatial autocorrelation analysis. OpenGeoDa is a very practical tool for spatial statistical analysis, which is often used in spatial autoregressive analysis [50].

The global Moran's *I* can be expressed by the following formula [51]:

$$I = \frac{n \sum_i \sum_{j \neq i} w_{ij}(y_i - \overline{y})}{\left(\sum_i \sum_{j \neq i} w_{ij}\right) \sum_i (y_i - \overline{y})^2} \tag{1}$$

where $w_{ij}$ represents the spatial weight matrix *W*, which corresponds to the geographic area (*i*, *j*), and $y_i$ and $y_j$ represent the observed values of *i* and *j*, respectively. The Moran's I value varies by [−1,1]. Positive values for Moran's *I* denote positive spatial autocorrelation; conversely, negative values show the negative; and 0 indicates no correlation. The absolute values of Moran's *I* are positively related to the spatial correlation of the selected variables.

Local autocorrelation analysis is usually used to identify hotspot areas by reflecting the correlation and similarity among adjacent units, which represents the state of PM$_{2.5}$ pollution gathering at spatial scale in this study. It could be expressed as [51]:

$$I = \frac{(x_i - \overline{x})}{\sum_i (x_i - \overline{x})^2} \sum_j w_{ij}(x_i - \overline{x}) \tag{2}$$

where $w_{ij}$ represents the spatial weight matrix $W$, which corresponds to the geographic area ($i$, $j$), and $y_i$ and $y_j$ represent the observed value of $i$ and $j$, respectively. In this paper, a Local Indicators of Spatial Association (LISA) agglomeration plot was used for analysis, which indicated the local spatial association of PM$_{2.5}$ pollution over space. There are four local spatial aggregation patterns on LISA agglomeration plot, namely the high–high type, the low–low type, the high–low type, and the low–high type. The high–high-type agglomeration and the low–low type agglomeration correspond to the positive spatial correlation in the global correlation index, while the low–high and high–low agglomeration correspond to the negative spatial correlation in the global correlation index.

### 2.3.2. Global Regression Model

The OLS model assumes that the variables are spatially stationary and it is usually utilized to quantitatively estimate the effects on a dependent variable, which is PM$_{2.5}$ concentration in this paper, caused by independent variables; however, if spatial correlation is ignored, it may lead to biased estimates. In order to solve this problem, the SLM and SEM were added in this paper. The SLM uses a lag variable to interpret the spatial autocorrelation, while SEM supplements the spatial correlation between different covariates that may be ignored through adopting error terms [52].

The OLS model could be displayed as:

$$y = \beta_0 + \sum_{j=1}^{p} \beta_j x_j + \varepsilon \tag{3}$$

where y means the PM$_{2.5}$ concentrations in sample cities, $\beta_0$ stands for the intercept, $\beta_j$ means the parameter estimate result for the selected explanatory variables $x_j$ ($j = 1, \ldots, p$), $p$ means the number of selected explanatory variables, and $\varepsilon$ stands for the error vector, which follows a normal distribution.

A variable can be related to the neighbors occasionally. So, based on the hypothesis that the dependent variables will change with the variation of the neighbors, the SLM is proposed. The SLM could be expressed as:

$$y = \beta_0 + \delta \sum_{j=1}^{p} W_j y + \sum_{j=1}^{p} \beta_j x_j + \varepsilon \tag{4}$$

where $\delta$ stands for the parameter estimate result of a spatial lag term, indicating the variation of spatial autocorrelation of y (when $\delta$ gets larger, the spatial autocorrelation of y gets higher); $W_j$ is a matrix of spatial weights of the dependent variable y. In this paper, we utilized the IDW as the method of spatial lag term weight.

When the OLS is applied to spatially related variables, the spatial correlation of the error terms is usually unavoidable, which contradicts the basic hypothesis of the OLS model. To solve the problem, the SEM resolves the impacts by the error term. The SEM model could be explained as:

$$y = \beta_0 + \sum_{j=1}^{p} \beta_j x_j + \mu \sum_{j=1}^{p} W_j' \varepsilon \tag{5}$$

where $\mu$ is the parameter estimate result of the error term, denoting the autocorrelation of the model's error term (when $\mu$ gets larger, the spatial autocorrelation of dependent variable gets higher); $W_j'$ is a matrix of spatial weights of the error term $\varepsilon$. To be consistent with the SLM, we utilized the IDW as the method of spatial error term weight.

In addition, the coefficient of determination and the Akaike information criterion (abbreviated as $R^2$ and AIC, respectively) were utilized in this paper to evaluate the simulation results of the global regression models. These two indices are often used to measure the fitting degree of various models; specifically, lower values of $R^2$ values and higher values of AIC represents not ideal models [53,54].

### 2.3.3. Local Regression Model

Although global regression models can describe the significance of statistical relationships between dependent and independent variables, it always results in problems by using global regression models because of the assumption that the variables have homogeneity. In other words, global regression models could not detect the potentially important local changes of the data [55]. To make up for this deficiency, the GWR model was added in this paper. The GWR model incorporates spatial location attributes of data into regression parameters. The GWR model uses local weighted ordinary least squares to estimate point parameters, whose weight is the distance function from the position of the regression point to the locations of other sample points. [56]. The GWR model could be explained as:

$$\ln y_j = \beta_{j0} + \sum_{j=1}^{n} \beta_{jn} x_{jn} + \varepsilon_j \tag{6}$$

where $j = 1, \ldots, 289$ represents spatial locations of the selected cities; $y_j$ represents the PM$_{2.5}$ concentration of the $j$ province (the dependent variable); six independent variables $x_{jn}$ ($n = 1, \ldots, 6$), including TR, PRE, WIN, SEC, HAILS, and VE; $\beta_{jn}$ are the local regression parameters; and $\varepsilon_j$ means the random error, by which each sample city in this paper has a set of corresponding parameters to detect the correlations between PM$_{2.5}$ pollution and related factors.

## 3. Results

### 3.1. Spatial Variation Characteristics of PM$_{2.5}$ Pollution and the Selected Explanatory Variables

Figure 3 reflects the spatial *variation* of PM$_{2.5}$ pollution in the selected 289 areas. In 2015, the annual average PM$_{2.5}$ concentration in the 289 sample cities was 56.6 μg/m$^3$ , and the five cities with the highest annual average concentration were Baoding, Xingtai, Hengshui, Dezhou, and Liaocheng. The annual average of the first four cities was greater than 100 μg/m$^3$ . According to (GB 3095-2012 Revision) the latest edition of the Ambient Air Quality Standard of the Ministry of National Environmental Protection of the People's Republic of China [45], the ambient air function is divided into two categories (Table 2). The level 1th standard is applicable to the special areas such as nature reserves and scenic sites, where the annual average and daily limits of PM$_{2.5}$ concentration are 15 μg/m$^3$ and 35 μg/m$^3$, respectively. The level 2th standard is applicable to residential, cultural, commercial, industrial, and rural areas, where the annual and daily limits are 35 μg/m$^3$ and 75 μg/m$^3$, respectively. In 2015, the annual average PM$_{2.5}$ concentrations of only 51 cities were lower than 35 μg/m$^3$; the annual average PM$_{2.5}$ concentrations of 209 cities were between 35 and 75 μg/m$^3$; the annual average PM$_{2.5}$ concentrations of 29 cities were higher than 75 μg/m$^3$.

**Table 2.** PM$_{2.5}$ concentration standards of China (μg/m$^3$ ).

| Concentration Limits | Annual Mean | 24-h Average |
|:---:|:---:|:---:|
| Level 1th | 15 | 35 |
| Level 2th | 35 | 75 |

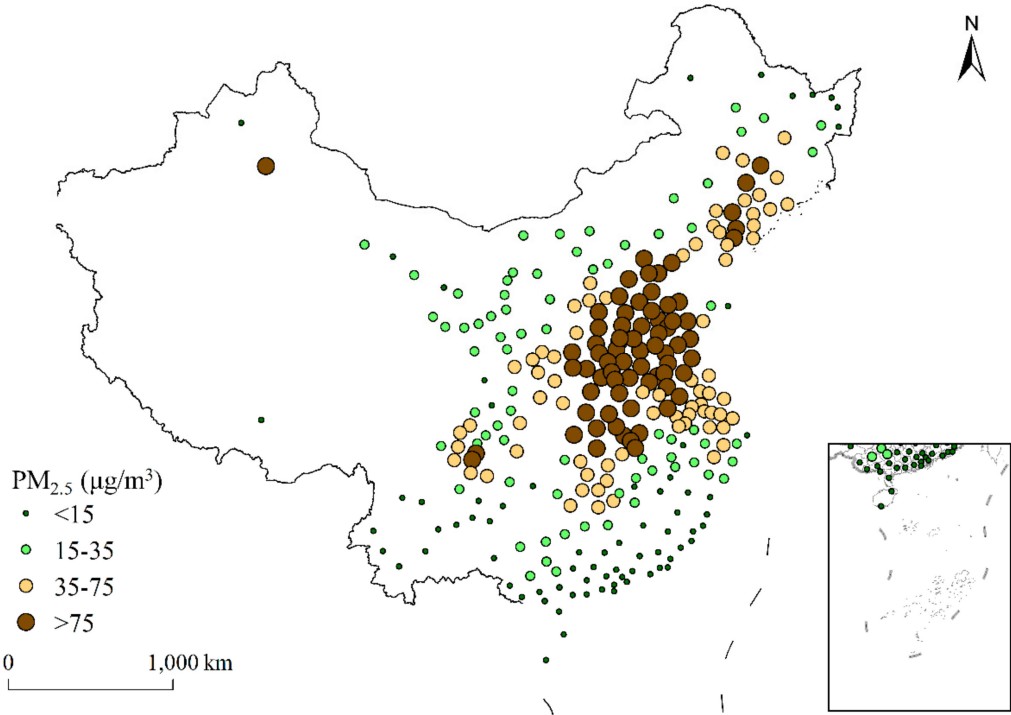

**Figure 3.** The spatial variation characteristics of PM$_{2.5}$ pollution in 289 cities in China in 2015.

Figure 4a reflects the spatial variation of TR in the study area. Overall, the eastern coastal cities have less terrain relief than the northwestern inland cities. Among them, the city with the lowest terrain relief was Yancheng, followed by Panjin and Nantong, and the city with the largest terrain relief was Weinan. Figure 4b reflects the spatial distribution of PRE in the study area. Huangshan (2524 mm) had the most abundant precipitation, while the most arid city was Jiayuguan (63 mm). Figure 4d reflects the development of secondary industry in each selected city. Panzhihua's SEC accounted for 71.45%, the highest proportion of all sample cities, and Heihe (15.17%) was the city with the lowest SEC. Figure 4e reflects the spatial distribution of HAILS of the selected cities. The city with the strongest HAILS was Shanghai (51.25%), followed by Beijing (49.47%), and the city with the lowest HAILS was Lhasa (5.7%). Figure 4f reflects the spatial distribution of VE in selected cities. Handan (94.1 million tons) had the highest emissions, followed by Chongqing (72.7 million tons), while Beihai (2.6 million tons) had the lowest emissions.

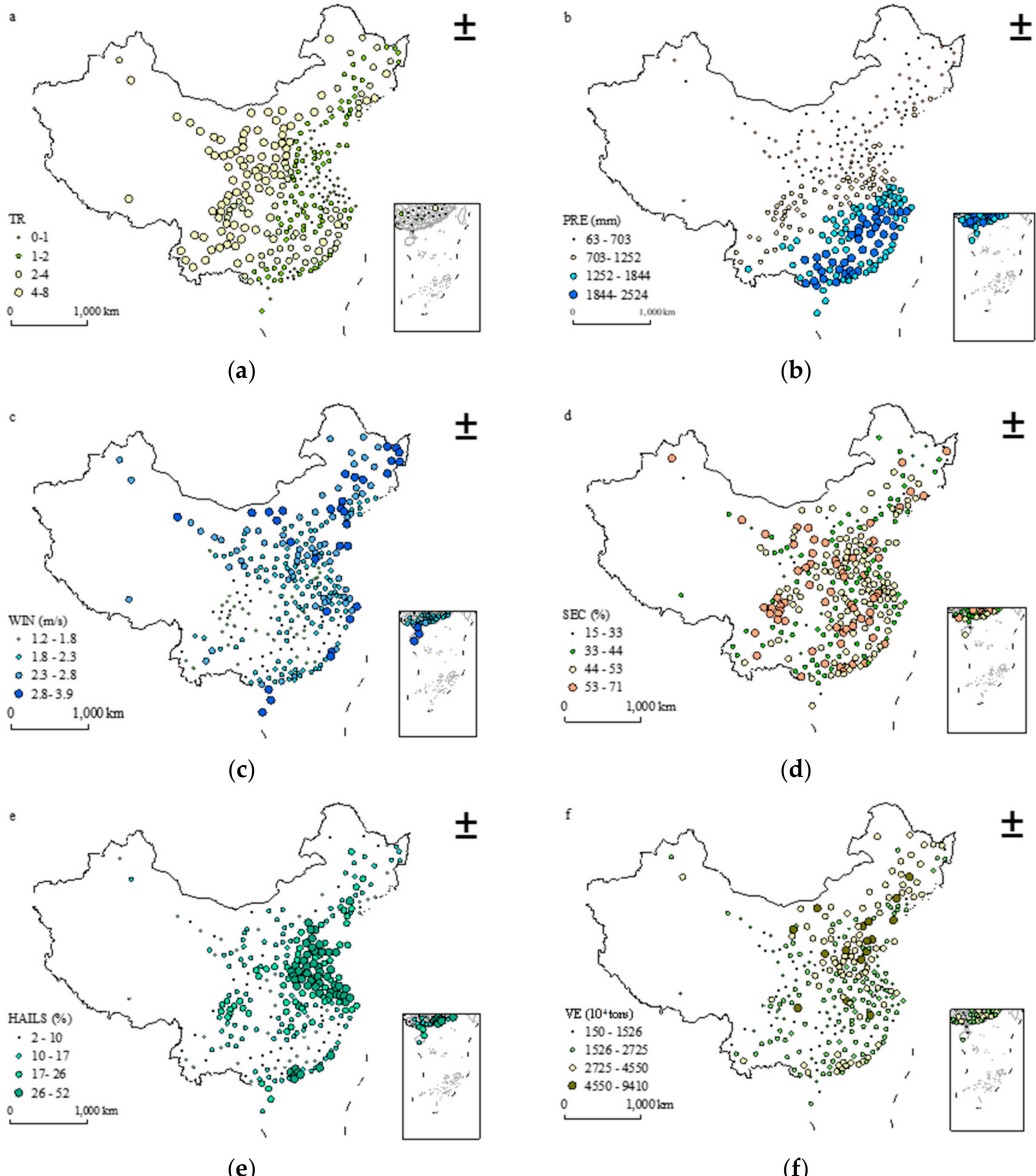

**Figure 4.** The spatial distribution characteristics of the (**a**) terrain relief (TR); (**b**) precipitation (PRE); (**c**) wind speed (WIN); (**d**) secondary industry's proportion (SEC); (**e**) human activity intensity of land surface (HAILS); and (**f**) total particulate matter emissions of motor vehicles (VE). The classification method used in Figure 4 and Figure 6 was the Natural Breaks (Jenks) method, which meant the classification criteria were based on natural groupings of data.

*3.2. Spatial Agglomeration of PM$_{2.5}$ Pollution Based on Moran's I*

The spatial autocorrelation test of the agglomeration characteristics of PM$_{2.5}$ pollution in selected cities was carried out using OpenGeoDa. By constructing the spatial distance weight matrix, the Moran's *I* value of PM$_{2.5}$ pollution in selected cities is 0.8406 (*p* < 0.01), which reflects significant spatial autocorrelation of PM$_{2.5}$ pollution in selected cities in 2015.

Figure 5 reveals that agglomeration characteristics of PM$_{2.5}$ pollution in China were particularly significant in the Beijing–Tianjin–Hebei region. The spatial distribution of PM$_{2.5}$ concentrations in Chinese cities in 2015 had evident spatial dependence and spatial spillover effects.

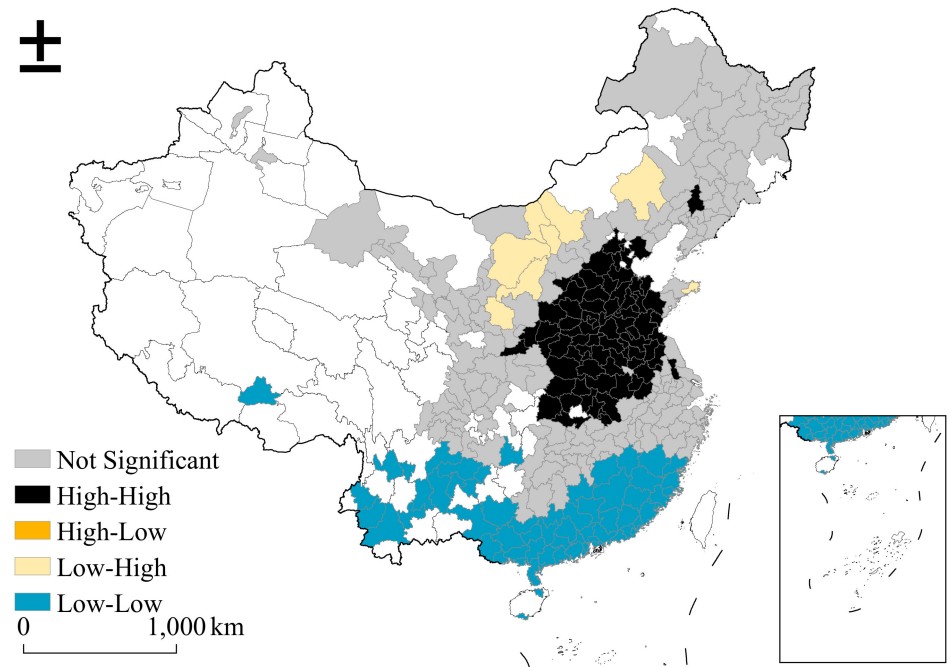

**Figure 5.** Moran's *I* Local Indicators of Spatial Association (LISA) agglomeration plot for PM$_{2.5}$ pollution in the study area.

### 3.3. Analysis of the Relationship Between PM$_{2.5}$ Pollution and the Explanatory Variables Based on Regression Models

Table 3 lists the estimated results of the relationship between PM$_{2.5}$ pollution and the explanatory variables based on three global regression models. The values of R$^2$ of the SLM model and the SEM model were 0.612 and 0.631, respectively, which were significantly higher than the result of the OLS model (0.513), indicating that the interpretation effect by using the SLM model and the SEM model was obviously better than using the OLS model. Furthermore, the SEM resulted in lesser AIC (1808.87) values than that of the SLM (1918.81), which meant the results of the SEM was better than the SLM. Taking the above into consideration, we chose the results of the SEM as the global regression analysis results.

**Table 3.** The correlation analysis between PM$_{2.5}$ concentrations and the selected explanatory variables based on global regression models (N = 289).

| Variables | OLS Results | SLM Results | SEM Results |
|---|---|---|---|
| TR | −1.51 | −0.81 | −0.41 |
| WIN (m/s) | −5.82 * | −1.5 ** | −4.14 * |
| SEC (%) | 0.25 ** | 0.07 *** | 0.06 *** |
| VE (10$^4$ tons) | 8.26 * | 5.52 ** | 3.85 ** |
| PRE (mm) | −0.017 ** | −0.031 *** | −0.04 *** |
| HAILS (%) | 0.67 * | 0.32 * | 0.27 ** |
| Lag parameter | – | 0.581 ** | – |
| Lambda (spatial error parameter) | – | – | 0.592 ** |
| R-squared | 0.513 | 0.612 | 0.631 |
| AIC | 2216.83 | 1918.81 | 1808.87 |

Note: *, **, and *** means that the significance is at the 5% level, 1% level, and 0.1% level, respectively.

From Table 3, we can see that no evident correlation existed between TR and PM$_{2.5}$ concentrations; while WIN and VE maintained the most significant correlation with PM$_{2.5}$ concentrations. WIN was obviously negatively correlated with PM$_{2.5}$ concentrations, namely, every 1 m/s increase in wind speed

could decrease $PM_{2.5}$ concentrations by 4.14 μg/m$^3$ . VE was specially positively correlated with $PM_{2.5}$ concentrations, which meant every 10,000 tons increase in VE could increase $PM_{2.5}$ concentration by 3.85 μg/m$^3$ . HAILS was positively correlated with $PM_{2.5}$ concentrations, denoting that every 1% rise in HAILS contributed to 0.27 μg/m$^3$ increase in $PM_{2.5}$ concentration. The correlation coefficient of PRE was −0.04, indicating a weak negative correlation between PRE and $PM_{2.5}$ concentration. For every 1 mm increase in precipitation, $PM_{2.5}$ concentration would reduce by 0.04 μg/m$^3$ . The coefficient of SEC was 0.06, indicating a weak positive correlation between SEC and $PM_{2.5}$ concentrations. For every 1% rise in SEC, the $PM_{2.5}$ concentration would increase by 0.06 μg/m$^3$ . In addition, the value of Lambda denoted an evident relationship with the $PM_{2.5}$ concentration, meaning that undiscovered spatial autocorrelation existed in the data.

The GWR model was then introduced to further analyze the local characteristics of the relation between $PM_{2.5}$ pollution and the selected explanatory variables. The $R^2$ of the GWR model was 0.734, which meant that 73.4% of the variability of the $PM_{2.5}$ pollution could be explained by the GWR model. Furthermore, the value of $R^2$ was higher than the estimation results obtained by the global models, which indicated the interpretation effectiveness of the GWR model behaved superior to the results of the OLS model, the SLM, and the SEM.

Figure 6a reveals the spatial distribution difference of local $R^2$, which increased from southwest to northeast (varying from 0.42 to 0.67). The variety of local $R^2$ suggested when comprehensively considering the effects of all the six explanatory variables, the spatially correlated relationship was more conclusively in northeastern cites than in other regions. At 0.67, Jiamusi had the largest local $R^2$ value, while Simao got the lowest local $R^2$ value of 0.42.

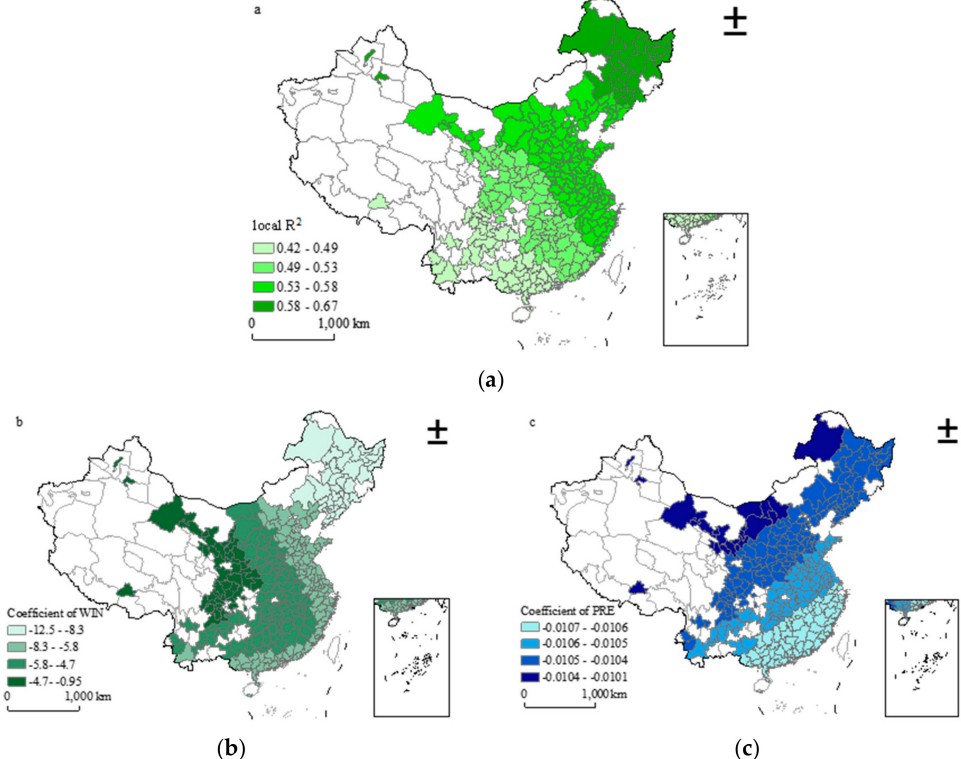

**(a)**

**(b)** **(c)**

**Figure 6.** *Cont.*

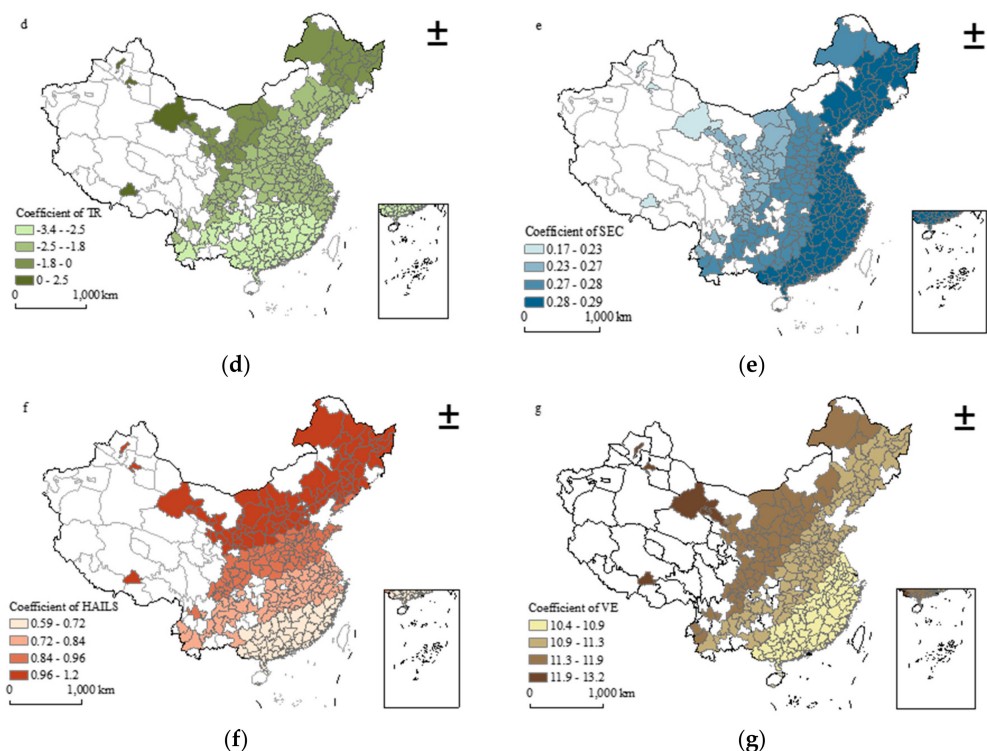

**Figure 6.** The results of the GWR model in 289 cities: (**a**) values of local R$^2$ and the coefficients of (**b**) WIN, (**c**) PRE, (**d**) TR, (**e**) SEC, (**f**) HAILS, and (**g**) VE, respectively.

Figure 6b reveals the spatial correlation between WIN and PM$_{2.5}$ pollution. The PM$_{2.5}$ concentration of Dalian had the strongest negative correlation with wind speed, and when the wind speed in Dalian increased by 1 m/s, the PM$_{2.5}$ concentration could be reduced by 12.5 μg/m$^3$ . Overall, there existed an evident negative relationship between WIN and PM$_{2.5}$ concentrations. The higher the wind speed, the lower the PM$_{2.5}$ concentration, and the faster the PM$_{2.5}$ pollution would be eliminated. In addition, it could be seen that from the eastern coast areas to the northwest inland areas, the effect of WIN on PM$_{2.5}$ pollution gradually weakened.

Figure 6c reveals the spatial correlation between PRE and PM$_{2.5}$ pollution. Guangzhou's PM$_{2.5}$ concentration showed the strongest negative correlation with PRE (−0.0107), which meant when precipitation in Guangzhou increased by 1 mm, the PM$_{2.5}$ concentration could be reduced by 0.0107 μg/m$^3$ . Similar to WIN, there was also a negative correlation between PRE and PM$_{2.5}$ concentrations, but the effect of PRE on PM$_{2.5}$ concentrations was significantly weaker. In addition, PRE in southeastern coastal areas was more beneficial to reducing PM$_{2.5}$ pollution than in the northwestern hinterland areas.

Figure 6d reveals the spatial correlation between TR and PM$_{2.5}$ pollution. Unlike the results of WIN and PRE, the spatial correlation between TR and PM$_{2.5}$ pollution was not spatially consistent. There existed a positive correlation between TR and PM$_{2.5}$ pollution in cities such as Urumqi (2.5) and Jiuquan (2.3) in the northwestern hinterland areas, indicating that the increase in TR in these areas would increase PM$_{2.5}$ concentrations. However, except for the few cities in the northwestern inland areas, we found negative correlations between TR and PM$_{2.5}$ concentrations in other areas. The PM$_{2.5}$ concentration in Haikou showed the strongest negative correlation (−3.4) with TR, which meant that when the TR of Haikou increased by 1, the PM$_{2.5}$ concentration could be reduced by 3.4 μg/m$^3$ .

Figure 6e reveals the spatial correlation between SEC and PM$_{2.5}$ pollution in the 289 selected cities. SEC showed a positively correlation with PM$_{2.5}$ pollution. Xiamen's PM$_{2.5}$ concentration had the strongest positive correlation with SEC (0.29), indicating that for every 1% rise in SEC in Xiamen, the PM$_{2.5}$ concentration would increase by 0.29 μg/m$^3$ .

Figure 6f reveals the spatial correlation between HAILS and $PM_{2.5}$ pollution. From south to north, the relationship between the two was gradually enhanced (varying from 0.59 to 1.2). The $PM_{2.5}$ concentration of Qiqihar had the strongest positive correlation with HAILS (1.2), indicating that the $PM_{2.5}$ concentration of Qiqihar would increase by 1.2 $\mu g/m^3$ for every 1% increase in HAILS.

Figure 6g reveals a significant positive correlation between VE and $PM_{2.5}$ pollution. Urumqi's $PM_{2.5}$ concentration was the strongest positive correlation with VE (13.2), indicating that $PM_{2.5}$ concentration would increase by 13.2 $\mu g/m^3$ for every 10,000 tons increase in VE in Urumqi. In addition, from the perspective of spatial variation trends, motor vehicle exhaust emissions in the northwest inland areas were more likely to cause $PM_{2.5}$ pollution than in the southeastern coastal areas.

## 4. Discussion

This study revealed that on the whole, PRE showed a negative relation to $PM_{2.5}$ pollution, while VE and SEC maintained a positive relation to $PM_{2.5}$ pollution—a finding that is generally consistent with that of related previous studies by Guo et al (2017) [57] and Wong et al (2019) [58].

Figure 7a shows that the value of the coefficient of PRE was reduced when the value of the coefficient of VE increased, which meant that in areas where VE was the critical factor leading to $PM_{2.5}$ pollution, the mitigation effect of PRE on $PM_{2.5}$ concentrations was weak. Similarly, Figure 7b suggests that in areas where VE was the critical factor leading to $PM_{2.5}$ pollution, the mitigation effect of SEC on $PM_{2.5}$ concentrations was weak. Therefore, in areas where the coefficient of VE was high, the impact of other natural environmental conditions and socio-economic factors such as PRE and SEC on $PM_{2.5}$ pollution would become weak.

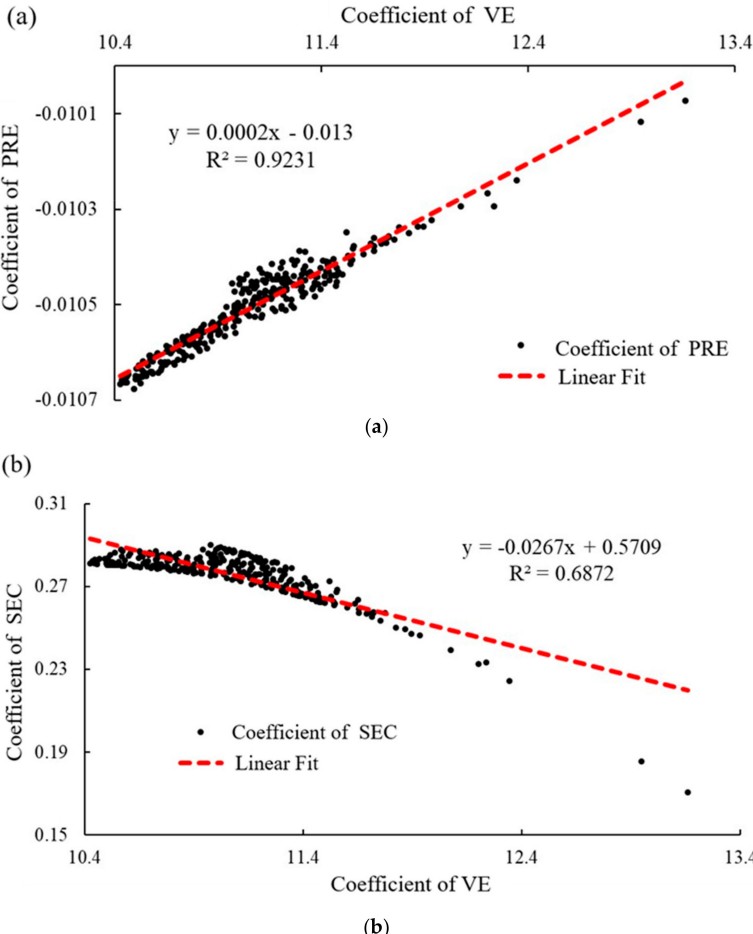

**Figure 7.** (**a**) The linear relationship between the coefficient of VE and coefficient of PRE; (**b**) the linear relationship between the coefficient of VE and the coefficient of SEC.

In brief, VE had the most significant effect on PM$_{2.5}$ concentrations among these related factors, which means controlling motor vehicle pollutant emissions is an important measure to reduce PM$_{2.5}$ pollution in China. As the largest developing country all over the world, China is suffering seriously from the damage caused by PM$_{2.5}$ pollution, which was linked with 0.964 million to 1.258 million deaths each year [59]. The sustainable development policy has been actively promoted by the Chinese government in recent years, which is plagued by severe PM$_{2.5}$ pollution. To achieve the goal of sustainable development, China needs to actively promote motor vehicles using clean energy.

## 5. Conclusions

Since 2013, the Chinese government has been vigorously advocating for PM$_{2.5}$ pollution control, but the results have not been significant. This paper analyzed the natural environmental conditions and the socio-economic factors related to the spatial variation of PM$_{2.5}$ concentrations based on a panel study of 289 cities. The conclusions drawn in this study are as follows.

First, the PM$_{2.5}$ concentrations in most cities in China exceeded their limits in 2015. The PM$_{2.5}$ pollution is not limited to a certain city or a certain area but is a nationwide problem. The average PM$_{2.5}$ concentration of the 289 selected cities in 2015 was 56.6 μg/m$^3$, and there were only 51 cities with PM$_{2.5}$ concentrations up to standard, accounting for 17.6% of the total sample cities. The spatial agglomeration characteristics of PM$_{2.5}$ pollution in China were particularly significant in the Beijing–Tianjin–Hebei region.

Second, the global regression models' results confirmed the significantly different spatial correlations between PM$_{2.5}$ pollution and the six selected explanatory variables. Specifically, there existed no significant correlation between TR and the PM$_{2.5}$ concentration; every 1 m/s increase in WIN could decrease the PM$_{2.5}$ concentration by 4.14 μg/m$^3$; every 10,000 tons increase in VE could increase the PM$_{2.5}$ concentration by 3.85 μg/m$^3$; the PM$_{2.5}$ concentration increased by 0.27 μg/m$^3$ with 1% increase in HAILS; every 1 mm increase in PRE would lead to a 0.04 μg/m$^3$ reduction of the PM$_{2.5}$ concentration; and every 1% rise in SEC could result in a 0.06 μg/m$^3$ rise in the PM$_{2.5}$ concentration.

Third, the simulation results of GWR l are better than those of the OLS model, the SLM, and the SEM model in explaining the local correlations between PM$_{2.5}$ concentrations and related factors. Overall, the effects of different explanatory variables on PM$_{2.5}$ concentrations varied greatly with geographical locations. When considering the comprehensive effects of all the six explanatory variables, the spatially correlated relationship was more conclusive in the northeastern cities than in other regions. VE had the greatest impact on PM$_{2.5}$ concentrations, and the impact intensity gradually increased from eastern coastal areas to the northwest inland areas. Among the three selected natural environmental conditions, WIN had the greatest impact on PM$_{2.5}$ concentrations, and the impact weakened from eastern coastal areas to northwest inland areas.

Fourth, the application of the GWR model improved the regression accuracy to a large extent by displaying fit characteristics compared with the results of the three global regression models.

What needs to be stated here is that the purpose of this paper is to detect the influence of related factors on spatial variations, rather than the temporal variations of PM$_{2.5}$ pollution. Therefore, the results can only be used to show the spatial variations of PM$_{2.5}$ pollution and cannot explain the temporal differences. This paper, which explored how the natural environmental conditions and socio-economic factors influence PM$_{2.5}$ pollution utilizing various regression models, provides a comprehensive perspective for further research.

**Author Contributions:** Conceptualization, S.Z.; Data curation, S.Z.; Formal analysis, S.Z.; Funding acquisition, Y.X.; Methodology, S.Z. and Y.X.; Project administration, Y.X.; Resources, Y.X.; Software, S.Z.; Supervision, Y.X.; Visualization, S.Z.; Writing—original draft, S.Z.; Writing—review and editing, S.Z. and Y.X.

**Funding:** This research was funded by the Strategic Priority Research Program of the Chinese Academy of Sciences, Grant No.XDA23020101 and China Scholarship Council, grant number 201804910734.

**Conflicts of Interest:** The authors declare no conflict of interest.

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
