# Peer review of "Exploring the Spatial Variation Characteristics and Influencing Factors of PM2.5 Pollution in China: Evidence from 289 Chinese Cities"

_sustainability, doi:10.3390/su11174751_

Round 1
Reviewer 1 Report
The objectives of this research are to describe the spatial variation of PM2.5 exposure and six explanatory variables (i.e. precipitation, terrain relief, wind speed, human activity intensity of land surface, the secondary industry’s proportion, and the total particulate matter emissions of motor) selected in 289 sample cities in China, to identify the relationship of spatial variation between PM2.5 concentrations and the six variables chosen, and to assess the strongest driving factors of spatial variation in PM2.5 exposure using global and local regression models. This research has the potential to be published in a journal. However, some revisions are required. Please consider my comments below.
Need attention related to technical writing (grammar; style of writing and so forth) Related to PM5 data, what time/year of data is used? (Line 106-107)
Is this the same as other variables? Please describe clearly (line 132) to facilitate identification, please add information “(a)” and “(b)” below the images For the sub-chapter 2.3.1 (line 141), I suppose that these descriptions have a clear reference. I suggest adding references in the description of this method (eg. getis, 1973) and please explain briefly, the main purpose of using this method in this study. “LISA” (line 155, 156, etc.), is it should be “Local Indicators of Spatial Association”. Please mention at least one explanation (line 163) what the meaning of “…….estimate the effects on a dependent variable caused by dependent variables”. Please revise if there is a writing error and briefly explain what the dependent variable is. Figure 3 (line 226-227), Because this study focused on the city scale, is it possible to display city boundaries in this generated map? Please consider and revise if needed I suggest to describe clearly the method/ geocode / software in the chapter "Data and Methods" (connecting with line 259) (line 284-287) to support the strength of the findings in this study and to respond the case in the results discussion part “why there existed no evident correlation between TR (terrain) and PM2.5? and etc.” I suggest that the author can explain some findings from previous studies that have similar results in the Discussion part.
Author Response
Dear reviewer,
Thank you so much for your comments!
Please see the attachment.

Reviewer 2 Report
The paper explores the characterization of the spatial distribution of Pm2.5 in China and analyses its potential factors with a wide range of classic spatial analysis and modelling techniques. It looks at both natural environmental and anthropological influences, of which the effect of HAILS is very interesting. Another innovation of the paper is to discuss the spatial varying mechanisms underlying the distribution of PM2.5. Some minor issues are as follow:
The authors may want to justify why only 289 cities are selected as the study area. There should references for the formulae of global and local Moran’s I Equation 3 is incorrect – the error term should be \epsilon_i (dropping the summation symbol); also please check all other equations are corrected expressedAuthor Response
Dear reviewer,
Thank you so much for your comments!
Please see the attachment.

This manuscript is a resubmission of an earlier submission. The following is a list of the peer review reports and author responses from that submission.